

# Correlation and mediation analysis between plasmapheresis donation behavior and bone mineral density and bone metabolism biomarkers: a cross-sectional study based on plasmapheresis donors at high risk of osteoporosis in China

Wan Li[1,*], Jia Liu[2,*], Changqing Li[1], Hui Yang[3], Yating Yang[4], Zhiwei Li[5], Shouqiang Yang[6], Yuan He[6], Guanglin Xiao[1], Ya Wang[1] and Yongjun Chen[1]

[1] Institute of Blood Transfusion, Chinese Academy of Medical Sciences & Peking Union Medical College, Chengdu, China
[2] West China Second University Hospital, SCU, Chengdu, China
[3] Nanyue Biopharmaceutical Corporation Ltd. China, Hengyang, China
[4] Grand Shuyang Life Sciences(Chengdu)Co., Ltd., Chengdu, China
[5] Linwu Plasmapheresis Station, Nanyue Biopharming Co., Ltd., Linwu, China
[6] Jiange Plasmapheresis Station, Sichuan Yuanda Shuyang Pharmaceutical Co., Ltd., Jiange, China
[*] These authors contributed equally to this work.

Corresponding authors
Ya Wang, 175235831@qq.com
Yongjun Chen, 563528422@qq.com

## ABSTRACT

**Background**. As a group of more than 3.67 million people, the bone health of Chinese plasmapheresis donors, which the main population is also a risk group of osteoporosis (OP), has raised concerns. Therefore, this article investigates the relationship between bone mineral density (BMD), bone metabolism indicators, and plasmapheresis donation behavior among some high-risk plasmapheresis donors for OP in China, and further explores the mediating factors through reasonable statistical methods.

**Methods**. Recruiting long-term and highly active plasmapheresis donors and new donors to measure the total calcium, albumin (ALB), 25-hydroxy vitamin D (25OHD), parathyroid hormone (PTH), type I procollagen amino-terminal peptide (P1NP), and type I collagen carboxy-terminal peptide ($\beta$-CTX) and serum ferritin (SF). Then, multiple linear regression was used to adjust confounding factors. Using restrictive cubic splines to explore the nonlinear relationship. Using the Bootstrap method, investigate whether SF has a mediating effect between plasmapheresis donation behavior and bone metabolism biomakers. Finally, analyze the differences in BMD between the two.

**Results**. Compared to new donors, repeat donors have a lower 25OHD, $\beta$-CTX and SF levels, while P1NP and PTH levels are high, with statistical differences. The Bootstrap analysis results show that SF level is a partial mediating factor between plasmapheresis donation behavior and bone metabolism biomakers, with a mediating effect ratio of 21.8%. There was no significant difference in the BMD between the two.

**Conclusion**. Long-term and frequently plasmapheresis donation does not affect the bone mass of even elderly donors at high risk for osteoporosis under the existing collection standards and anticoagulant use in China. However, as a self-regulation way, it does increase the osteogenic activity of the body.

# INTRODUCTION

Osteoporosis (OP) is an age-increasing disease with osteopenia and destruction of bone microstructure, and its serious consequence is osteoporotic fracture (*Su et al., 2024*). According to the results of the 2018 China Epidemiological Survey on Osteoporosis, osteoporosis has become an important health problem for people over 50 years old in China, with a prevalence rate of 19.2% and a low bone mass rate of 46.4% (*National Health Commission of the People's Republic of China, 2018*). According to China's Seventh Population Census, China has the largest elderly population in the world (*Chen, 2023*; *Zhen, 2023*). As the population ages, these rates will rise rapidly. At present, the pathogenesis of osteoporosis is not fully understood, but aging and menopause are two important influencing factors of osteoporosis (*Lane, Russell & Khan, 2000*). Therefore, there is an urgent need to focus on bone health in people over the age of 50.

Generally, the indicators of bone metabolism in blood will be tested clinically to evaluate the status of bone metabolism, and can be used as a short-term monitoring tool to evaluate the efficacy of OP. The level of bone turnover markers (BTMs) is usually used to measure the strength of osteogenesis and osteoclastic activity. BTMs include bone formation markers (*e.g.*, P1NP, *etc.*) and bone resorption markers (*e.g.*, type I collagen carboxy-terminal peptide ($\beta$-CTX), *etc.*) (*Zhen, 2023*).

Plasmapheresis is the collection of plasma components by centrifugation of whole blood in an extracorporeal circuit, and the remaining blood components are returned to the donor (*Okafor et al., 2010*). Citrate is the standard anticoagulant used in the plasmapheresis process (*Lee & Arepally, 2012*). In China, the concentration of anticoagulant used and the ratio to whole blood were 4% and 1:16, with a dose of approximately 70–90 ml per donor per time. Acute side effects of citrate (*e.g.*, perioral numbness and even hypocalcemia) have been reported (*Lee & Arepally, 2012*; *Hegde et al., 2016*; *Sigler, Lee & Srivaths, 2018*). After acute citrate infusion, the average decrease in calcium ion levels in the body can reach 25%–35% (*Hester et al., 1983*), and in severe cases, hypocalcemia may occur. After citrate is chelated with calcium, it is excreted in the urine. Therefore, it can cause partial calcium loss with urine. Skeletons are the body's calcium reservoir, storing 99% of the body's calcium (*Peacock, 2010*). When the body encounters a decrease in serum calcium levels, parathyroid hormone (PTH) will be secreted in a timely manner and act on multiple organs in the body, including bones. An increase in PTH enhances the activity of osteoclasts in the bone, thereby releasing calcium ions into the bloodstream to maintain blood calcium

levels (*Chen et al., 2011*). Overall, it is to maintain blood calcium levels by mobilizing calcium in the bones. However, it is currently unknown whether long-term and frequent decreases in serum calcium levels will affect bone health. At present, the results of studies on the effect of repeated plasmapheresis donors on bone are inconsistent. *Amrein et al. (2010)* found that the apheresis group had a higher rate of low bone mass compared with the non-apheresis group, and there was a statistically significant difference in lumbar BMD between the two groups. However, a prospective study by *Bialkowski et al. (2019)* showed that apheresis did not cause a significant difference in BMD. The frequency of donation, volume of anticoagulant used per time vary between countries (*National Health Commission of the People's Republic of China, 2022*). These findings do not necessarily reflect the status of the Chinese donors. The 46-60-year-old group accounted for 84.4% of the total plasmapheresis population (*Sun et al., 2021*). With the growth of plasmapheresis donor population, the number of women donor in China is now about twice that of men. There is a high degree of overlap between this population and those at high risk of osteoporosis. There's no study to date on the bone health of plasmapheresis donors in China.

In recent years, an increasing number of studies have suggested a close relationship between iron metabolism and bone metabolism (*Wang, 2018*; *Cheng et al., 2019*). In the preliminary research of our research group, we found that plasmapheresis has two different health effects on iron metabolism. Plasma donation increased the risk of iron deficiency in women, while alleviated the risk of iron accumulation in men. Therefore, apheresis may also have a certain correlation with bone metabolism and BMD through its impact on iron metabolism.

Therefore, this article investigates the relationship between bone mineral density (BMD), bone metabolism indicators, and plasmapheresis donation behavior among some high-risk plasmapheresis donors for OP in China, and further explores the mediating factors through reasonable statistical methods, providing a certain theoretical basis for ensuring the health and safety of Chinese plasmapheresis donors.

## METHODS

### Study design and participants

This is a multicenter, cross-sectional study. From July 1, 2022 to September 30, 2022, plasmapheresis stations in Linwu (Hunan Province) and Jiange (Sichuan Province) recruited plasma donors as research subjects (the specific process of including and excluding subjects can be found in the Fig. S1). In order to minimize the interference of both primary and secondary osteoporosis risk factors as confounding factors, we used the International Osteoporosis Foundation (IOF) Osteoporosis Risk 1-Minute Test Questions for donor screening to ensure that no donor already exists the risk of osteoporosis. This questionnaire is recommended as a commonly used tool for osteoporosis risk assessment in the Guidelines for the Diagnosis and Management of Primary Osteoporosis 2017 (*Xia et al., 2019*).

Inclusion criteria: (1) Registered qualified plasmapheresis donors (repeat donors,we named this group donors) who have donated more than 50 times and have continued

to donate in the past six months, as well as new donors (we named this group controls) who have no history of plasmapheresis donation; (2) Male donors are over 50 years old, and female donors are postmenopausal; (3) Complete "the International Osteoporosis Foundation (IOF) Osteoporosis Risk 1-Minute Test Questions", all answers are no; (4) Obtain the consent of the plasmapheresis donor and sign the "Informed Consent Form of the Plasmapheresis Donor". Exclusion criteria: (1) Donors who may be affected by X-rays are excluded; (2) Taking steroid hormones or drugs that affect BMD and bone metabolism in the past 3 months should be excluded.

This study passed the review of the Ethics Committee of the Institute of Blood Transfusion, Chinese Academy of Medical Sciences (approval number: NO: 2022029), and had each subject sign an informed consent form.

### Sample size estimation

This article uses a cross-sectional study sample size calculation formula for estimation ($Wang \& Cheng, 2020$), and the formula is as follows where p represents the expected prevalence of osteoporosis in the donor population. Since there is no survey on the prevalence of osteoporosis in the donor population, and the average age of the population surveyed in this survey is over 50 years old. Therefore, using the results of the 2018 China Osteoporosis Epidemiological Survey in people over 50 years old ($National Health Commission of the People's Republic of China, 2018$), it can be seen that $p$ is 19.2%. $q = 1 - p$, $d$ is the allowable error, and z is the significance test statistic. Set $\alpha = 0.05$, $d = 0.2 \times p$, $z = 1.96$. According to the formula, the sample size needs to be at least 421 people.

$$n = \frac{pq}{(d/z_\alpha)^2} = \frac{z_\alpha^2 \times pq}{d^2}.$$

### Blood sampling and information collection

A 3 ml venous blood was collected from each donor in a vacuum blood collection tube with a separating gel before donating plasma. Centrifugation (3,000r/min, 10min) was used to separate the serum, and the serum was stored at low temperature in a $-20\,^\circ C$ freezer, and finally detected in the same lab. Blood test items include: serum total calcium, albumin (ALB), 25OHD, PTH, P1NP, $\beta$-CTX, SF levels.

Indicators of plasmapheresis donation behaviors including the total number of plasmapheresis donations, the recent plasma donation frequency (*i.e.,* the number of plasmapheresis donations in the last 12 months) and the interval from the last plasma donation were retrieved from the information system of the plasmapheresis station. Self-made questionnaires were used to collect background information such as age, BMI, annual household income, meat intake, calcium supplementation, and vitamin D supplementation. The International Physical Activity Questionnaire (IPAQ) was used to collect the physical activity of plasmapheresis donors.

### Laboratory analysis and assays

P1NP was detected by total type I collagen amino-terminal elongated peptide detection kit (batch number: 653011, Roche, Basel, Switzerland), and the detection instrument was

Roche E602. PTH was detected using a parathyroid hormone assay kit (batch number: 220901, Mindray, Shenzhen) and the testing instrument was Mindray CL-6000i. $\beta$-CTX was detected with $\beta$-collagen special sequence detection kit (batch number: 635899, Roche, Basel, Switzerland), and the detection instrument was Roche E602. Albumin was detected using an albumin detection kit (batch number: AUZ1125, Beckman, Brea, CA, USA) and the detection instrument was Beckman 5800. Serum calcium was detected by calcium assay kit (batch number: AUZ1199, Beckman, Brea, CA, USA), and the detection instrument was Beckman 5800. 25OHD was detected by total 25-hydroxyvitamin D assay kit (batch number: 230501, Mindray, Shenzhen), and the detection instrument was Mindray CL-6000i. SF was detected using chemiluminescence method using a ferritin assay kit (batch number: 47177UDOO, Abbott, Chicago, IL, USA) and the detection instrument was Abbott Alinity.

DXA was used to measure the BMD of anterior and posterior lumbar spine (L1-L4), bilateral femoral neck and non-dominant radial BMD of plasmapheresis donors with GE LunarDPX Prodigy. In both Linwu and Jiange plasmapheresis stations, Haier Blood Technology XJ-II blood component apheresis machine was used for plasmapheresis donation.

## Study outcome parameters

P1NP was measured by electrochemiluminescence with normal reference values ranging from 16.3 to 78.2 ng/mL, and data were cited from the Guidelines for Clinical Application of Bone Turnover Biomarkers (*Li, Xia & Zhang, 2021*). $\beta$-CTX was measured by electrochemiluminescence, and the normal reference value ranged from 0.114–0.628 ng/mL, and the data were quoted from the 2021 Guidelines for the Clinical Use of Biochemical Markers for Osteotransition. Albumin was determined by BCG, with normal reference values ranging from 40 to 55 g/L, and data were quoted from the *Hong, Yusan & Ziyu (2014)*. Serum calcium was determined by colorimetric method, and the normal reference value ranged from 2.2–2.7 mmol/L, and the data were quoted from the "National Clinical Laboratory Operating Regulations". PTH was determined by chemiluminescence method, and the normal reference value ranged from 12-65 pg/mL, and the data were quoted from the "National Clinical Laboratory Operating Regulations". 25OHD was determined by electrochemiluminescence, and vitamin D inadequacy was indicated at <20 ng/ml, as shown in the Guidelines for the Use of Vitamin D and Bone Health in Adults (*Liao et al., 2014*). According to the WHO criteria for iron accumulation (*World Health Organization, (2020)*), male SF > 200 ug/L and female SF > 150 ug/L are classified as iron accumulation. BMD is presented as g/cm$^2$ and as $T$-score. Based on the DXA classification criteria for BMD, a $T$-value of $\geq$-1 is normal, a $T$-value of $-2.5 < T$ <-1 is low bone mass, and a $T$-value of $\leq-2.5$ is osteoporosis, as shown in the Guidelines for the Diagnosis and Management of Primary Osteoporosis 2017 (*Xia et al., 2019*). According to the guideline, a $T$-value $\leq-2.5$ at either site is diagnosed as osteoporosis.

## Statistical analysis

Graphpad Prism 8.0 and SPSS 24.0 (Armonk, NY, USA) were used for data collation and statistical analysis. Firstly, the overall population was descriptively statistically analyzed,

and the basic differences between the two groups of new plasmapheresis donors and repeat plasmapheresis donors were compared. For the continuous data, the data conforming to the normal distribution were expressed as mean and standard deviation, and one-way ANOVA was used. Data that do not conform to the normal distribution are expressed by median and interquartile ranges, and Wilcoxon rank-sum test is used. The frequency (composition ratio) was used to express the counting data, and the chi-square test was used. The overall population was divided into groups according to the type of donor, and whether there were statistically significant differences between groups in bone biomarkers. The statistically significant indexes were analyzed according to three different indicators of donation behaviors (*i.e.,* the total number of plasmapheresis donations, recent frequency, and the interval from the last plasma donation), and multiple linear regression analysis was used to further assess the association between donation behaviors and bone metabolism biomarkers. We use Restricted Cubic Splines (RCS) to better demonstrate the nonlinear trend of change between bone metabolism biomakers and plasmapheresis donation behaviors. The Bootstrap method was used to test the mediating effect of SF on bone metabolism markers in relation to plasma donation behaviors. Finally, analyze whether there are statistically significant differences in BMD at different locations among different types of plasmapheresis donors (new and repeated donors). When conducting multiple comparisons, the Bonferroni method is used to adjust the $P$-value.

The test level was $\alpha = 0.05$, and $p < 0.05$ indicated that the difference was statistically significant.

## Quality control

In the information collection stage, a unified and standardized project operation manual and questionnaire were produced and the staffs involved in the project were trained through online meetings to ensure the standardization of data collection. In the data entry stage, two independent members entered separately and then cross-checked to ensure the accuracy of the data. In the statistical analysis stage, the detection value of serum total calcium was corrected by the correction formula [corrected calcium = measured calcium + (40-measured albumin) × 0.02] and included in the statistical analysis.

## RESULTS

### Baseline characteristics

A total of 553 donors were included in this study, including 276 new donors (*i.e.,* the control group) and 277 repeat donors (*i.e.,* the donor group), respectively. There were statistically significant differences in age, annual household income, exercise intensity, protein intake and calcium supplementation between the two groups ($P < 0.05$), as shown in Table 1. There were significant differences in the levels of 25OHD, P1NP, $\beta$-CTX and PTH between the two groups ($P < 0.05$). The inadequacy rate of vitamin D in the repeat donor group (82.31%) was significantly higher than that in the new donor group (68.48%) ($P < 0.001$).

**Table 1  Basic characteristics.**

|  |  | Controls | Donors | *P*-value |
|---|---|---|---|---|
| *N* | | 276 | 277 | – |
| Total number of donation | | – | 107 (54.5,196.5) | – |
| Questionnaire information | | | | |
| Sex male (*n*, %) | | 177 (63.9%) | 100 (36.1%) | 0.77 |
| Age (year, mean ± SD) | | 52.74 ± 2.13 | 53.64 ± 2.78 | <0.001 |
| BMI (kg/m 2, mean ± SD) | | 25.73 ± 5.50 | 25.39 ± 3.40 | 0.38 |
| Annual household income | Low | 8 (2.90%) | 41 (14.80%) | |
| | Medium | 78 (28.30%) | 199 (71.80%) | |
| Physical activity (IPAQ) | Low | 64 (23.20%) | 52 (18.80%) | |
| | Medium | 174 (63.00%) | 221 (79.80%) | |
| Protein intake | Rarely | 25 (9.10%) | 2 (0.70%) | |
| | Often | 44 (15.90%) | 175 (63.20%) | |
| Calcium supplementation | Rarely | 258 (93.50%) | 270 (97.50%) | |
| | Often | 18 (6.50%) | 7 (2.50%) | |
| Vitamin D supplementation | Rarely | 276 (100.00%) | 277 (100.00%) | |
| | Often | 0 (0.00%) | 0 (0.00%) | |
| Bone metabolism indicators | | | | |
| 25OHD (ng/mL) | | 17.24 ± 8.37 | 13.02 ± 7.09 | <0.001 |
| P1NP (ng/mL) | | 64.31 ± 24.16 | 70.98 ± 44.94 | 0.03 |
| $\beta$-CTX (ng/mL) | | 0.34 ± 0.20 | 0.30 ± 0.15 | <0.001 |
| PTH (pg/mL) | | 40.73 ± 17.72 | 49.01 ± 22.02 | <0.001 |
| Total Ca (mmol/L) | | 2.33 ± 0.08 | 2.34 ± 0.06 | 0.62 |

## Effects of different donation behavior indicators on bone metabolism biomarkers

In this part of the study, the statistically different indicators related to bone metabolism biomarkers in Table 1 was grouped according to three donation behavior indicators (*i.e.,* total number of donations, the recent frequency of donation, and interval between the last donation). The results showed that the level of 25OHD and $\beta$-CTX was lower while PTH and PINP were higher in the repeat donor group. High total number of donations, high recent frequency and short interval have significant influence on these biomarkers. After 100 times of plasmapheresis donation in Fig. 1, it has an impact on the bone formation indicators (P1NP). When the frequency of plasmapheresis donation increases to more than 17 times recently, both bone resorption ($\beta$-CTX) and bone formation (P1NP) indicators undergo significant changes. When the interval between plasmapheresis donation exceeds 14 days, it has no effect on the levels of bone resorption ($\beta$-CTX) and bone formation (P1NP) indicators. When the interval between plasmapheresis donation exceeds 28 days, the levels of 25OHD and PTH return to before plasmapheresis donation. In summary, the main influencing factors of bone metabolism related indicators are recent plasmapheresis donation frequency. When it increases to a certain level (*i.e.,* more than 17 times/nearly 12 months, note: the current upper limit in China is 24 times/nearly 12 months), both

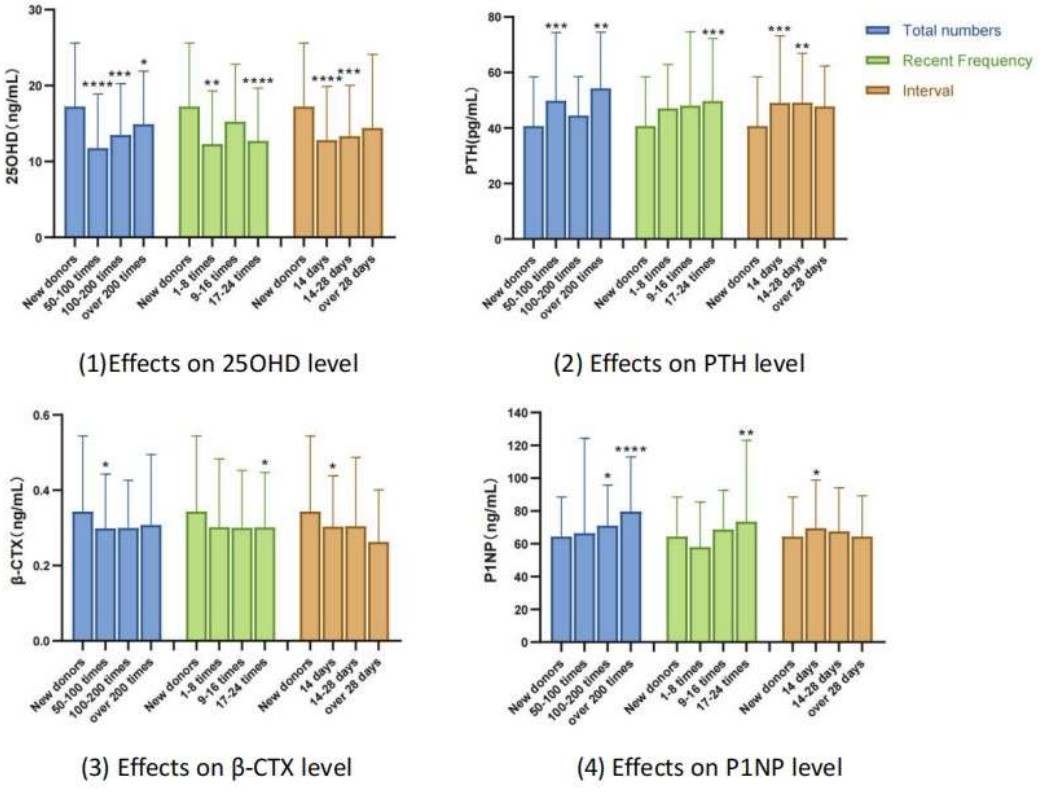

**Figure 1 Effects of three different indicators of plasmapheresis donation behaviors on bone biomarkers in donors.** *$P < 0.05$; **$P < 0.01$; ***$P < 0.001$; ****$P < 0.0001$.

bone resorption and bone formation indicators undergo significant changes. When the interval between plasmapheresis donation exceeds 14 days (the current minimum interval requirement in China), the levels of bone resorption and bone formation indicators return to before plasmapheresis donation. When the interval between plasmapheresis donation exceeds 28 days, the levels of 25OHD and PTH return to before plasmapheresis donation. As the total number of plasmapheresis donation increases, it first affects bone resorption indicators, and after 100 plasmapheresis donations, it will further affect bone formation indicators.

## Multiple linear regression analysis on bone metabolism biomarkers

Multiple linear regression analysis was performed for bone metabolism biomarkers (25OHD, P1NP, $\beta$-CTX, PTH) with significant differences in Table 1, and adjusted for a large number of possible confounding factors (age, body mass index (BMI), annual household income, physical activity, protein intake, calcium supplementation and vitamin D supplementation). Only indicators of plasmapheresis donation behaviors that were clearly risk factors were shown (Table 2). The results showed that the level of 25OHD was positively correlated with the total number of plasmapheresis donations, and negatively correlated with the frequency of plasmapheresis donation. The level of P1NP was positively

**Table 2 Multiple linear regression on bone metabolism biomarkers.**

| Dependent variable | Indicators of plasmapheresis donation behaviors | Unstandardized coefficients | | Standardized coefficients beta | t | Sig. | Collinearity statistics | |
|---|---|---|---|---|---|---|---|---|
| | | B | Std. error | | | | Tolerence | VIF |
| 25OHD | Total numbers | 0.021 | 0.005 | 0.239 | 4.458 | <0.001 | 0.38 | 2.63 |
| | Recent frequency | −0.29 | 0.046 | −0.386 | −6.367 | <0.001 | 0.297 | 3.365 |
| P1NP | Total numbers | 0.036 | 0.022 | 0.093 | 1.663 | 0.037 | 0.534 | 1.874 |
| $\beta$-CTX | Recent frequency | −0.002 | 0.001 | −0.122 | −2.336 | 0.02 | 0.635 | 1.576 |
| PTH | Recent frequency | 0.368 | 0.14 | 0.185 | 2.621 | 0.009 | 0.544 | 1.839 |

correlated with the total number of donations, the higher the number of donations, the higher the level of P1NP. PTH level was positively correlated with the frequency of donation, while $\beta$-CTX level was negatively correlated with the frequency of donation. The interval from the last plasma donation has no correlation with changes in the indicators after adjustment.

## Nonlinear trends between indicators of plasmapheresis donation behaviors and bone metabolism biomarkers

From Fig. 2, it can be seen that non-linear relationships existed between biomarkers and behavior indicators in Table 2. The 25OHD level decreased rapidly at first with the increase of the total numbers, then began to rise towards 100 times, and then remained stable after 200 times. However, it decreased rapidly with the increase of recent frequency. It may be noteworthy that even new donors have vitamin D inadequacies, which may be exacerbated by plasma donation. Both PTH and P1NP levels showed an upward trend as the total number and recent frequency increased. $\beta$-CTX decreased slowly as the recent frequency decreased, but values remained within the normal range.

## Baseline comparison and mediation analysis of SF

The results indicate that there is a statistical difference between the two groups at the SF level. Compared with the control group, the donor group had lower SF levels. The iron accumulation rate in the group of women who repeatedly donate plasma is significantly lower than that in the group of new donors ($P < 0.001$). There is no statistically significant difference in iron accumulation rate between male repeat donors and new donors (Table S1).

Total number of donations, the recent frequency of donation, interval between the last donation, SF, $\beta$-CTX and P1NP were included in the Pearsoncorrelation analysis to analyze the correlation between the two pairs. There is a correlation between total numbers and the levels of SF and P1NP, with correlation coefficients of −0.306 and 0.148, respectively. The level of SF is correlated with total numbers and P1NP, with correlation coefficients of −0.306 and −0.187, respectively. P1NP is correlated with total numbers and SF (Table S2). The above results indicate that there is a correlation between total numbers, SF, and P1NP pairwise. Therefore, total numbers, SF, and P1NP were selected for the next analysis.

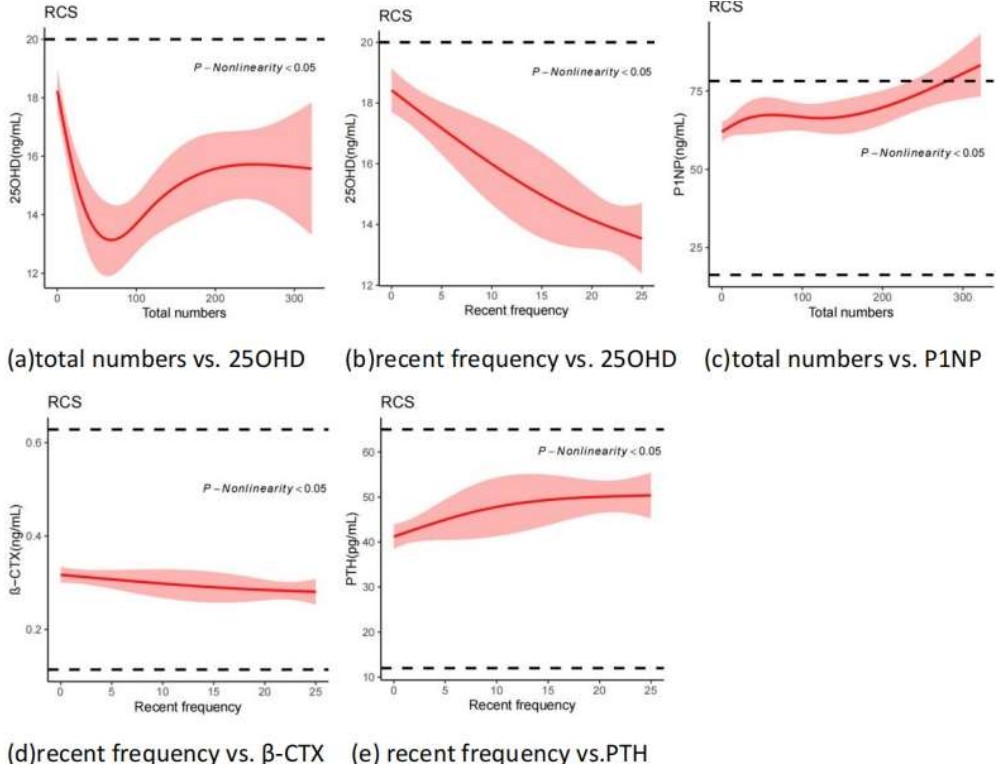

**Figure 2** Nonlinear correlation between behaviors of plasmapheresis donation and bonemetabolism biomarkers (Restricted Cubic Splines).

Use Bootstrap method for mediating effect analysis. According to the research purpose of this article, the variable details are as follows: total numbers is the independent variable, SF level is the mediator variable, and P1NP level is the dependent variable, as shown in Fig. 3. According to the results in Table 3, it can be concluded that SF partially mediates the relationship between the total number of plasma donations and the level of P1NP. The total effect is $c = 0.156$, the direct effect is $c' = 0.122$, and the indirect effect is $a * b = 0.034$. The mediating effect percentage is $a*b/c = 21.8\%$.

## Analysis related to BMD

There was no significant difference in BMD between the two groups ($P > 0.05$). There was no difference in the prevalence of osteoporosis and low bone mass rate between the two groups, indicating that there was no significant difference in BMD between the two groups (Table 4). Meanwhile, we conducted multiple linear regression analysis on BMD in different parts and found that plasmapheresis donation related behaviors were not independent factors affecting BMD (Table S3), which is consistent with the single factor results. Therefore, this indicates that plasmapheresis donation does not affect the donor's BMD.

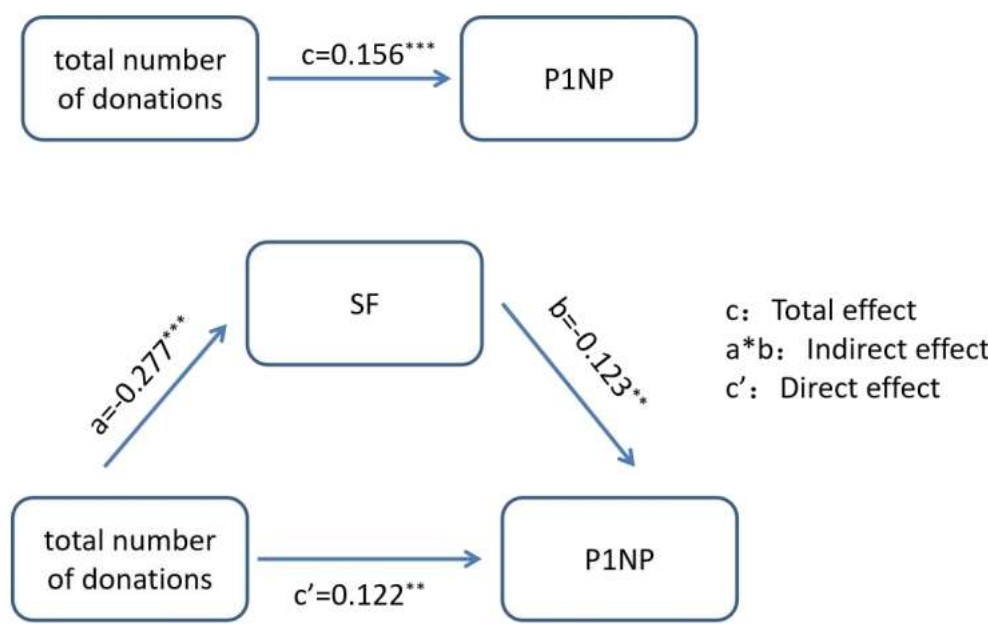

**Figure 3** The mediating effect diagram between the total number of donations, SF, and P1NP. ** $P <$ 0.01; *** $P < 0.001$.

**Table 3** Regression analysis of SF mediation model (standardized).

| Predictive variables | Model 1 | | Model 2 | | Model 3 | |
|---|---|---|---|---|---|---|
| | $\beta$ | $t$ | $\beta$ | $t$ | $\beta$ | $t$ |
| Total number of donations | −0.277 | −6.774*** | 0.156 | 3.722*** | 0.122 | 2.812*** |
| SF | | | | | −0.123 | −2.838** |
| $R^2$ | 0.076 | | 0.024 | | 0.038 | |
| $F$ | 45.899 | | 13.859 | | 11.045 | |

Notes.

Model 1, Predicts the SF level based on the total number of donations; Model 2, Prediction of P1NP level based on the total number of donations; Model 3, Predicting P1NP level together with the total number of donations and SF level.

*$P < 0.05$.
**$P < 0.01$.
***$P < 0.001$.

# DISCUSSION

Menopause and aging are two important causes of osteoporosis (*ACOG Committee on Clinical Practice Guidelines—Gynecology*). The peak bone mass of women is inherently lower than that of men, and coupled with the physiological menopause causing a cliff like decline in estrogen levels, postmenopausal women experience faster bone loss and more severe osteoporosis (*Yuan et al., 2019*). At present, there have been a few studies on both short-term and long-term effects of plasmapheresis on bone metabolism (*Amrein et al., 2010*; *Sun et al., 2019*; *Dao et al., 2011*; *Chu et al., 2010*; *Bialkowski et al., 2016*), but little attention has been paid on those at high risk for osteoporosis. There are also a few studies on Chinese whole blood donor and platelet apheresis donors, but none on plasmapheresis

**Table 4 Comparison of two groups of BMD and abnormal rates.**

|  | Controls | Donors | *P*-value |
|---|---|---|---|
| Total | 276 | 277 | – |
| BMD-related indicators |  |  |  |
| Lumbar spine L1–L4 BMD (g/cm 2) | $0.96 \pm 0.15$ | $0.99 \pm 0.13$ | 0.20 |
| Lumbar spine T-score | $-1.9 \pm 1.2$ | $-1.73 \pm 1.04$ | 0.31 |
| Left femoral neck BMD (g/cm 2) | $0.83 \pm 0.2$ | $0.87 \pm 0.13$ | 0.06 |
| Left femoral neck T-score | $-1.48 \pm 0.95$ | $-1.26 \pm 0.98$ | 0.11 |
| Right femoral neck BMD (g/cm 2) | $0.83 \pm 0.14$ | $0.86 \pm 0.19$ | 0.35 |
| Right femoral neck T-score | $-1.43 \pm 0.99$ | $-1.29 \pm 1.00$ | 0.34 |
| Radius BMD (g/cm 2) | $0.56 \pm 0.23$ | $0.55 \pm 0.25$ | 0.77 |
| Radius T-score | $-2.44 \pm -2.67$ | $-2.67 \pm 2.01$ | 0.28 |
| BMD-related abnormal rates |  |  |  |
| Prevalence of osteoporosis | 47.46% (131/276) | 50.54% (140/277) | 0.469 |
| Low bone mass rate | 32.97% (91/276) | 32.13% (89/277) | 0.833 |

donors. These results do indeed indicate that exposure to citrate during the apheresis procedure acutely affects mineral and bone metabolism. But long-term effects on BMD are different. As a group of more than 3.67 million people (*People's Medical Publishing House, 2019*), the bone health of Chinese plasmapheresis donors, which the main population is also a risk group of osteoporosis, has raised concerns.

This article is the first survey in China to focus on the long-term effects of plasmapheresis on bone metabolism and BMD in high-risk plasmapheresis donors for osteoporosis, namely postmenopausal women and men over 50 years old. Bone turnover biomarkers (BTM) are products of bone tissue's own metabolism, representing the dynamics of systemic bone metabolism (*Schini et al., 2023*). BTM includes bone formation (such as P1NP, osteocalcin) and bone resorption (such as $\beta$-CTX) (*Eastell & Szulc, 2017*). The dynamic balance between osteogenic activity and osteoclast activity is the foundation for maintaining stable bone mass in the body. The bone mass is also regulated by various hormones. Among them, vitamin D, as one of the important hormones for regulating bones, can increase the absorption of calcium in the intestine and increase osteogenic activity. Currently, 25OHD is a good indicator of vitamin D levels in the body (*Delrue & Speeckaert, 2023*). As one of the three major hormones regulating calcium and phosphorus, PTH plays an important role in the regulation of bone metabolism. In our study, there was no statistically significant difference in BMD at all measurement sites between the donors group and the control group. However, 25OHD and $\beta$-CTX levels were significantly lower, PTH and P1NP levels were significantly higher in donor group ($p < 0.05$).

The trend of changes in 25OHD, PTH, and P1NP is consistent with the research of *Amrein et al. (2010)* and *Chen et al. (2011)*. But interestingly, the level of $\beta$-CTX shows the opposite. The decrease in 25OHD promotes an increase in PTH levels. Previous studies have reported that increasing level of vitamin D can decrease secretion of PTH in the body (*Babić Leko et al., 2021*) and intermittent administration of PTH can promote osteogenic activity in the body (*Li et al., 2019*). Similarly, *Boot et al. (2015)* investigated

postmenopausal female apheresis donors and found that the BMD of the lumbar spine and hip joints in the donor group was higher than that in the whole blood donor group. The authors analyzed that this may be due to the chronic long-term elevation of PTH caused bone resorption in a long-term effect. Additionally, in our study, the interval from the last plasma donation has no correlation with changes in the indicators after adjusting for confounding factors. The result is consistent with the research of *Amrein et al. (2010)*, in which most of the bone metabolic indicators returned to normal within one day.

For bone metabolism results, it can be seen that the main influencing factor of bone metabolism related indexes was the frequency of recent plasma donation. When the frequency of recent plasma donation increased to a certain extent (*i.e.,* more than 17 times/nearly 12 months, note: the current upper limit in China is 24 times/nearly 12 months), the bone resorption and bone formation indicators changed significantly. When the interval between plasma donation was longer than 14 days (the current minimum interval in China), the bone resorption and osteogenesis indicators returned to their baseline levels. When the interval between plasma donations exceeded 28 days, the levels of 25OHD and PTH returned to those before donation. With the increase of the total times of plasma donation, the bone resorption indicators were affected first, and the bone formation indicators were further affected after 100 times of plasma donation.These finds indicated that plasma donation behavior and usage of citrate, to a certain extent, promoted osteogenesis and reduce osteoclast of the body. This may be a stress and self-regulation way for the body to cope with the imbalance of calcium homeostasis and calcium loss, thereby ensuring bone health.

Further, a high rate of vitamin D inadequacy was observed in our study,, similar to national epidemiological data (*Liao et al., 2014*). Multiple linear regression analysis demonstrated a significant negative correlation between 25OHD levels and recent donation frequency, consistent with Fig. 2B. The main form of vitamin D in the blood is its binding to $\alpha$ globulins (*Khazai, Judd & Tangpricha, 2008*). Due to the loss of plasma protein caused by donation, 25OHD levels reduced as a result. It can also be seen from Figs. 1A and 1B that the level of 25OHD was inversely proportional to the change in PTH. Therefore, it is recommended to provide extensive health education to plasmapheresis donors, and long-term plasmapheresis donors should increase their outdoor activity time and supplement vitamin D appropriately since plasma donation can further increase vitamin D deficiency.

The theory of iron accumulation is one of the mechanisms underlying osteoporosis in recent years. Clinical research observations have shown that the BMD of iron accumulating individuals is lower than that of the general population, and there is a statistically significant difference between the two. Using the Bootstrap method to study the mediating effect, it was found that SF levels partially mediate the relationship between plasma donation behavior and P1NP, with a mediating effect ratio of 21.8%. This suggests that plasma donors can directly affect P1NP through their plasma donation behavior, and can indirectly affect P1NP by affecting SF levels. So we made a hypothesis that plasma donation might improve iron accumulation for individuals at high risk of osteoporosis, thereby improving bone

metabolism and alleviating the progression of osteoporosis due to SF's effect of mediation. However, no evidence was found in our study.

Since plasma donation was found to be significantly associated with changes in bone metabolism parameters in our study, we had high expectations for its further effect on BMD. However, no significant difference in BMD found finally between the two groups. Our results are consistent with the studies of Bialkowski et al. (2019), Sun et al. (2019), Xu et al. (2012), that is, plasmapheresis will not damage the BMD of donors. A prospective study by Grau et al. (2017) also showed that frequent plasmapheresis donation does not increase the risk of fractures. However, Amrein et al. (2010) found that compared to the control group, the lumbar spine BMD in the donor group was lower with statistical differences. This study cannot exclude the influence of genetic, nutritional, or lifestyle risk factors on the results, and there is heterogeneity in donation frequency among the study population, with over 80% of researchers rarely participating in donations (less than once a month).

The above findings, including ours, indicated that several underlying mechanisms may have contributed alone or acted in concert to maintain stable bone mass for donors: repetitive stress to calcium homeostasis induced by citrate exposure, and/or recurring biological factor changes including specific factors potentially important for bone and mineral homeostasis (such as chronic long-term elevation of PTH). The use of anticoagulant in the process of collection and saline infusion for hemodilution may cause hypocalcemia and short-term alterations in bone metabolism, but fortunately, the body can basically return to normal levels in a short time. However, after long-term and frequent donation, even elderly donors at high risk for osteoporosis may cause self-regulation of the body to promote osteogenesis and reduce osteoclast, and ultimately maintaining stable bone mass of donors despite vitamin D level reduction (Fig. 4).

This study recruited long-term and highly active plasmapheresis donors at high risk for osteoporosis as study participants (i.e., postmenopausal women and men over 50 years old, donation times ranged 54.5–196.5), and explored the possible effects of plasma donation behavior indicators on bone metabolism biomarkers from three dimensions: long-term, medium-term and short-term. We also used IOF Osteoporosis Risk 1-Minute Test Questions and appropriate statistical methods to minimize the effect of background confounding. These are complementary and innovative to previous similar studies. However, this article still has the following shortcomings. (1) This article only investigates two plasmapheresis stations and cannot comprehensively reflect the overall situation of the Chinese plasmapheresis population. Further multicenter and larger sample studies are needed. (2) This article is a cross-sectional study, which can only preliminarily explore the correlation between plasmapheresis donation and bone metabolism markers, and cannot conduct further mechanism research, and cannot strictly control the confounding factors, and the evidence for causal inference is low, so further prospective research is needed.

In summary, long-term and frequent plasmapheresis donation does not affect the bone mass of even elderly donors at high risk for osteoporosis under the existing collection standards and anticoagulant use in China. However, as a self-regulator way, it does increase the osteogenic activity of the body. It also recommended that middle-aged and elderly plasmapheresis donors should be appropriately supplemented with vitamin D.

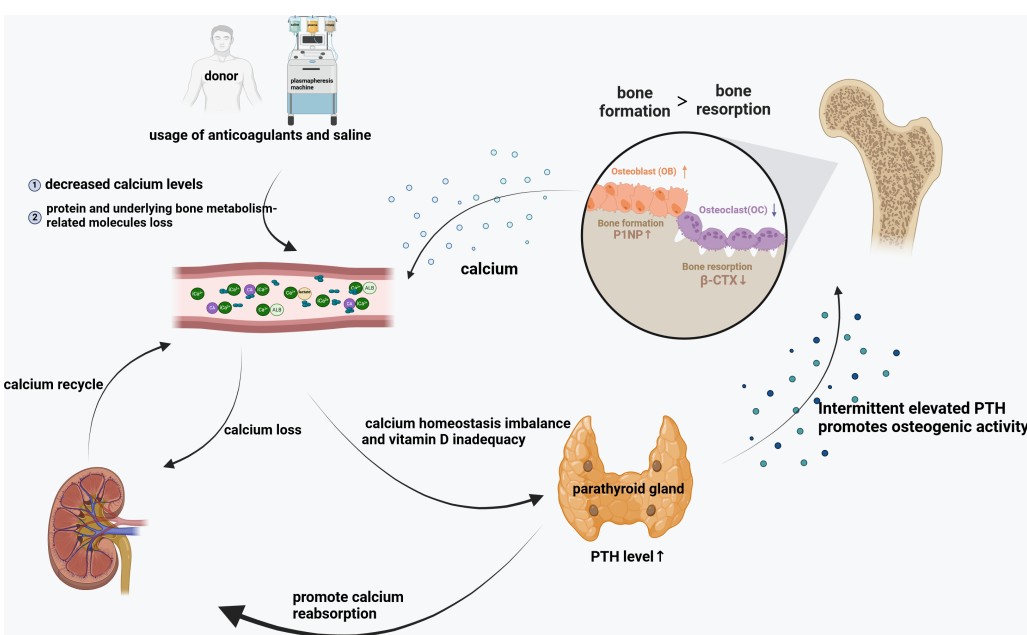

**Figure 4** Bone metabolism changes after long-term and frequent plasmapheresis donation.

## ACKNOWLEDGEMENTS

Thanks to You QingQing for the support and encouragement.

### Funding

This work was supported by CAMS Innovation Fund for Medical Sciences (CIFMS), Grant Number: 2021-I2M-1-060. The funders had no role in study design, data collection and analysis, decision to publish, or preparation of the manuscript.

### Grant Disclosures

The following grant information was disclosed by the authors:
CAMS Innovation Fund for Medical Sciences (CIFMS): 2021-I2M-1-060.

### Competing Interests

Hui Yang & Zhiwei Li are employed by Nanyue Biopharming Co., Ltd. and Yating Yang, Shouqiang Yang & Yuan He are employed by Sichuan Yuanda Shuyang Pharmaceutical Co., Ltd. They declare the following on the conflict of interest: They promise to avoid conflicts of interest (even superficial conflicts) with the company, its shareholders and its customers.

### Author Contributions

- Wan Li analyzed the data, prepared figures and/or tables, authored or reviewed drafts of the article, and approved the final draft.

- Jia Liu conceived and designed the experiments, analyzed the data, prepared figures and/or tables, authored or reviewed drafts of the article, and approved the final draft.
- Changqing Li performed the experiments, authored or reviewed drafts of the article, and approved the final draft.
- Hui Yang performed the experiments, authored or reviewed drafts of the article, and approved the final draft.
- Yating Yang performed the experiments, authored or reviewed drafts of the article, and approved the final draft.
- Zhiwei Li performed the experiments, authored or reviewed drafts of the article, and approved the final draft.
- Shouqiang Yang performed the experiments, authored or reviewed drafts of the article, and approved the final draft.
- Yuan He performed the experiments, authored or reviewed drafts of the article, and approved the final draft.
- Guanglin Xiao conceived and designed the experiments, analyzed the data, authored or reviewed drafts of the article, and approved the final draft.
- Ya Wang conceived and designed the experiments, authored or reviewed drafts of the article, and approved the final draft.
- Yongjun Chen conceived and designed the experiments, authored or reviewed drafts of the article, and approved the final draft.

### Human Ethics

The following information was supplied relating to ethical approvals (*i.e.*, approving body and any reference numbers):

This study passed the review of the Ethics Committee of the Institute of Blood Transfusion, Chinese Academy of Medical Sciences (approval number: NO: 2022029).

### Data Availability

The raw measurements are available in the Supplementary Files.

### Supplemental Information

Supplemental information for this article can be found online at http://dx.doi.org/10.7717/peerj.18589#supplemental-information.

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
