# Peer review of "Correlation and mediation analysis between plasmapheresis donation behavior and bone mineral density and bone metabolism biomarkers: a cross-sectional study based on plasmapheresis donors at high risk of osteoporosis in China"

_PeerJ, doi:10.7717/peerj.18589_

## Round 0.1 · original submission · Major Revisions

Although two of the three reviewers recommended rejection, I have decided to give you an opportunity to rebut and revise manuscript accordingly. Please address all the issues pointed by the reviewers and provide revised manuscript.

·

Basic reporting

The introduction and methods were well written and comprehensive! The study design was clear and replicable, and each parameter thoroughly described and well referenced. Furthermore, all raw data was provided under the Supplemental section. The suggestions may be as follows:

1) Figure 1 is not mentioned until later in the manuscript, and there's no mention of it in the Results section. Please discuss Figure 1 under the Results section and have figures numbered in its respective order.

2) "RCS results" from line 307 is not defined in the manuscript before being mentioned under Discussion.

3) I suggest to describe the bone biomarker levels in relation to donor group as "levels were significantly lower" and " levels were significantly higher" in line 277 to underscore your findings.

4) English was clear but had a few spelling, grammar, and punctuation/spacing errors; such as in Line 220 ("existed"), 235 ("Pearson"), 323 ("No evidence WAS found in our study"), 343 ("vitamin D level reduction"), 335 ("frequent plasmapheresis" not frequently), 357 ("self-regulator"); as well as extra comas in line 305, and missing first capital letters after a period (i.e. 263, 71,..)

Experimental design

Research question was well defined and methods described in detail. Further comment found in the other sections.

Validity of the findings

Discussion and conclusions are well stated indicating the novelty of the study and bringing the focus of the study back to the original question. The effect of plasmapheresis on donors' bone mass density (BMD) and osteoporosis (OP) hasn't been studied before. Unlike other studies that have looked at apheresis and whole blood donors, this new study is looking at plasmapheresis donors and their bone turnover biomarkers, concluding that plasmapheresis does not have a significant impact on BMP and OP.

Additional comments

No further comment.

Reviewer 2 ·

Basic reporting

.

Experimental design

.

Validity of the findings

.

Additional comments

This article is the first survey in China to focus on the long-term effects of plasmapheresis on bone 267 metabolism and BMD in high-risk plasmapheresis donors for osteoporosis, namely postmenopausal women
268 and men over 50 years old.By addressing the following points, the manuscript may be strengthened:
1. While the study compares new and repeat plasmapheresis donors, it need to include a non-donor control group to better understand the unique effects of plasmapheresis on bone metabolism independent of other potential confounders.

2. Given the high rate of vitamin D inadequacy observed in the study, it would be valuable to explore this relationship further. This could include assessing dietary intake, sunlight exposure, and the impact of vitamin D supplementation on bone metabolism in plasmapheresis donors.

3. The manuscript should provide more detailed information on the statistical methods used, including the handling of missing data, the assumptions underlying the statistical tests, and the adjustment for multiple comparisons. Additionally, the use of advanced statistical techniques, such as mixed-effects models, could account for the repeated measures and donor variability.

4. While the study identifies associations between plasmapheresis donation and bone metabolism markers, it lacks a deep dive into the underlying biological mechanisms. Further research into how citrate and calcium homeostasis directly impact bone cells and molecular pathways would be valuable.

5. The discussion section should expand on the clinical implications of the findings, including how they might influence guidelines for plasmapheresis donors, especially those at high risk for osteoporosis.

Reviewer 3 ·

Basic reporting

The English in the manuscript requires substantial revision, and the article needs to be restructured.

Experimental design

The aim of this study is to investigate the relationship between bone mineral density (BMD), bone metabolism indicators, and plasmapheresis donation behavior among high-risk plasmapheresis donors for osteoporosis (OP) in China. A multiple regression analysis on BMD should also be performed.

The presentation of the tables was unclear, particularly the meaning of the controls and the donors.

Table 2 was confusing as well, especially regarding the difference between the B value and the beta coefficient.

Validity of the findings

The statement 'The results showed that the level of 25OHD was positively correlated with the total number of plasmapheresis donations and negatively correlated with the frequency of plasmapheresis donation' is confusing. You should define what you mean by the 'frequency' of plasmapheresis donation.

Additional comments

None

---

## Round 0.2 · Minor Revisions

Please address remaining issues pointed by the reviewer and revise manuscript accordingly.

·

Basic reporting

The manuscript has been much improved from the first submission. Except for some small proofreading, the English is also much better. The only other suggestion I might have is to define or write out the full names of the acronyms, including the blood tests, i.e., SF (serum ferritin), and the formula under the Methods section, as well as the titles under the Results section.
Also, another minor suggestion is to move some of the explanations on the blood tests and what they determine under the Discussion section into the Introduction/Background section.

Experimental design

-

Validity of the findings

-

Reviewer 3 ·

Basic reporting

Ok, no more comments

Experimental design

Ok no more comments

Validity of the findings

Ok

Additional comments

No more

---

## Round 0.3 · accepted · Accept

All remaining concerns were adequately addressed and revised manuscript is acceptable now.